# Fully Convolutional One-Stage 3D Object Detection on LiDAR Range Images

**Zhi Tian**[1], **Xiangxiang Chu**[1], **Xiaoming Wang**[2]*, **Xiaolin Wei**[1], **Chunhua Shen**[3†]

[1] Meituan Inc.    [2] Northwestern Polytechnical University    [3] Zhejiang University

[1] {tianzhi02,chuxiangxiang,weixiaolin02}@meituan.com
[2] chunhua@me.com   [3] xiaomingwang80@163.com

## Abstract

We present a simple yet effective fully convolutional one-stage 3D object detector for LiDAR point clouds of autonomous driving scenes, termed FCOS-LiDAR. Unlike the dominant methods that use the bird-eye view (BEV), our proposed detector detects objects from the range view (RV, *a.k.a.* range image) of the LiDAR points. Due to the range view's compactness and compatibility with the LiDAR sensors' sampling process on self-driving cars, the range view-based object detector can be realized by solely exploiting the vanilla 2D convolutions, departing from the BEV-based methods which often involve complicated voxelization operations and sparse convolutions.

For the first time, we show that an RV-based 3D detector with standard 2D convolutions alone can achieve comparable performance to state-of-the-art BEV-based detectors while being significantly faster and simpler. More importantly, almost all previous range view-based detectors only focus on single-frame point clouds, since it is challenging to fuse multi-frame point clouds into a single range view. In this work, we tackle this challenging issue with a novel range view projection mechanism, and for the first time demonstrate the benefits of fusing multi-frame point clouds for a range-view based detector. Extensive experiments on nuScenes show the superiority of our proposed method and we believe that our work can be strong evidence that an RV-based 3D detector can compare favourably with the current mainstream BEV-based detectors. Code will be made publicly available.

## 1 Introduction

With the rise of autonomous driving, 3D object detection from the LiDAR point cloud has been recently drawing increasing attention. Similar to the 2D image object detection [RHGS15, TSCH19, LAE$^+$16, RF17, RDGF16], 3D object detection requires the model to predict the (3D) locations of the objects of interest and the associated properties (*e.g.*, categories, sizes, heading, and the state of motion). In spite of the unprecedented success that the computer vision community has attained on the 2D image object detection, it is still intractable to transfer the success to the 3D object detection task.

Most previous 3D object detection methods consider that the point cloud is amorphous and consists of a set of unordered points. Thus, this task is considered significantly different from its 2D detection counterpart, which works on structured RGB images. Moreover, the cornerstones of modern computer vision—CNNs or CNN-like vision transformers (*e.g.*, Swin Transformers [LLC$^+$21]) also assume

---

*Work done as an intern at Meituan Inc.

†Corresponding author.

36th Conference on Neural Information Processing Systems (NeurIPS 2022).

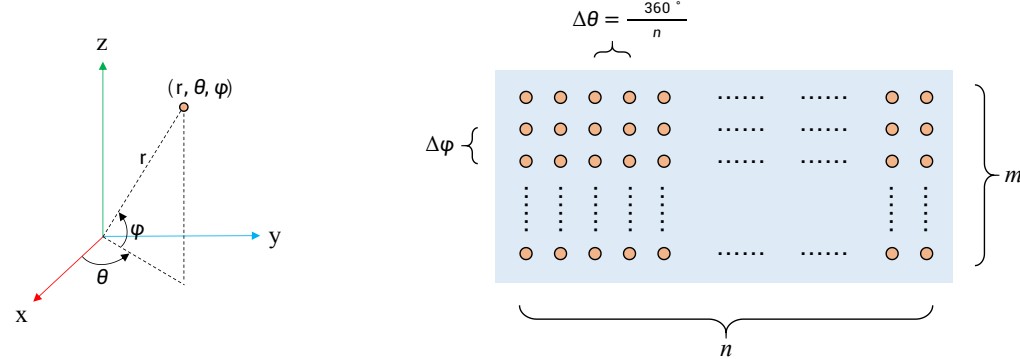

Figure 1: **The range image of the point cloud** (right). The size of the range image is $m \times n$, where $m$ is the number of beams and $n$ is the number of measurements (*i.e.*, sampling frequency) per scan cycle. A 3D point's coordinates on the range image are computed by discretizing the azimuthal angle $\theta$ and the inclination angle $\phi$ in the spherical coordinate system (left). The $m$, $n$, and $\Delta\phi$ depend on the LiDAR specifications.

that the inputs are well-organized as grids, which poses another difficulty of accurate 3D object detection in point clouds. As a result, in order to adopt these well-developed techniques, almost all top-performing point cloud object detection methods first partition the 3D space into *structured* voxels (or pillars) [ZT18, YZK21, YML18, LVC$^+$19, Li17] and then follow a paradigm similar to that of 2D object detectors [RHGS15, ZWK19]. Additionally, given the prior that it is very rare two objects being stacked along with the elevation axis in autonomous driving scenes, most methods only carry out the detection task on the bird-eye view (BEV) of the point cloud, which can reduce the exponentially increasing complexity resulted from the third dimension. However, owing to incompatibility with the LiDAR's sampling process in the autonomous driving scenes, BEV-based solutions often suffer from the following shortcomings. 1) In fact, the points in autonomous driving are regularly sampled in the spherical coordinate system with the origin being the LiDAR sensor. The BEV disregards this regularity, and causes the issue that the voxels far away from the origin have much fewer points than the ones near the origin; and a large number of voxels are even empty. This results in the need for sparse convolutions [YML18, YZK21, Gra14], significantly complicating the system, particularly for on-device applications. 2) The points in a voxel have to be sampled or padded so that every voxel has the same number of points. The sampling decimates a large number of points, leading to the loss of information before the model see anything. 3) From the BEV, some objects such as "pedestrian" and "traffic cone" become very small. Accurately detecting these objects requires a fine-grained voxel size, dramatically increasing the price of computation.

In this work, we advocate a new solution for 3D object detection that works on the range view (RV). As noted by many previous works [FXW$^+$21, SKD$^+$20, MLK$^+$19], if we consider these points in the spherical coordinate system and project them in terms of their inclination and azimuth angles, they can form a compact 2D image with size being $m \times n$ (shown in Fig. 1), where $m$ is the number of beams (*i.e.*, channels) of the LiDAR sensor and $n$ is the sampling frequency per scan cycle. The resulting image is referred to as "range image" or "range view" of the point cloud. Compared to the aforementioned BEV, the range image is nearly dense and compatible with the LiDAR sampling process, eliminating the need for sparse convolutions and alleviating the loss of points. In addition, the range image closely resembles the common RGB image, minimizing the cost of transferring the 2D detection methods to 3D ones. In the literature, some works attempted to detect objects on the range view such as RangeDet [FXW$^+$21] and LaserNet [MLK$^+$19]. These works have shown that RV-based methods can also achieve decent detection performance, showing the promise. However, previous RV-based methods only focus on the single-frame point cloud as the aforementioned range view structure does not hold anymore if the ego (and the origin) moves between the multiple frames. Additionally, the range image of a single-frame point cloud is already nearly dense so that there are not many vacancies the points from other frames can populate. These issues make the range-view based detectors difficult to benefit from the multi-frame fusion. In sharp contrast, the multi-frame fusion can dramatically improve the performance in the BEV-based

detectors, as shown in [YZK21, LVC$^+$19]. This makes the performance of RV-based detectors largely lag behind that of the BEV-based ones, hampering their development and application. In this work, we show this issue can be largely remedied with a well-designed Multi-round Range View (MRV) projection mechanism. The proposed MRV makes the RV-based detectors be able to enjoy the gain of multi-frame fusion and thus achieve competitive performance with multi-frame BEV-based detectors.

Here, we summarize our main contributions as follows.

- We propose a fully convolutional one-stage 3D object detector, termed FCOS-LiDAR. FCOS-LiDAR works on the LiDAR range images and minimizes the gap between the 3D and 2D detectors, while being substantially simpler than current mainstream BEV-based 3D detectors [YZK21, ZT18].

- Compared to previous BEV-based detectors [ZT18, YZK21], FCOS-LiDAR sidesteps the complicated voxelization process and eliminates the need for sparse convolutions due to the highly compact range view representation of the point cloud. In the setting of only using the single-frame point cloud as inputs, FCOS-LiDAR can outperform the state-of-the-art BEV-based detector CenterPoint [YZK21] while being faster.

- We also present a well-designed Multi-round Range View (MRV) projection mechanism, making RV-based detectors be able to benefit from the multi-frame fusion of point clouds as well, and achieve competitive performance compared to the multi-frame BEV-based detectors. To our knowledge, we are the first one approaching the challenge of the RV-based multi-frame point clouds fusion and showing that RV-based detectors can also be boosted by the multi-frame fusion of point clouds.

- We believe that our excellent performance of the RV-based detector can be a strong evidence that RV-based detectors compare favorably against the mainstream BEV-based detectors and encourage the community to pay attention to this promising solution.

## 2   Related Work

**Bird-view based 3D Detection.** Most top-performing LiDAR-based 3D detectors [FPZ$^+$21, YML18, YZK21, LVC$^+$19, ZT18] fall into this category, which first convert the point cloud into BEV images. VoxelNet [ZT18] is the first end-to-end BEV-based detector, which employs PointNet [QSMG17] to handle the representation within a voxel and 3D convolutions to generate high-level features for the region proposal network (RPN). SECOND [YML18] proposes to use sparse convolutions, which can save the computing burden of 3D convolutions. Another popular approach is to eliminate the voxelization along the elevation axis and convert the point cloud into the pillars [LVC$^+$19]. Based on the voxel-based or pillar-based BEV representation, CenterPoint [YZK21] achieves state-of-the-art performance by using the anchor-free pipelines.

**Range-view based 3D Detection.** Due to the compactness of the RV representation, some methods [SWC$^+$21, BSM$^+$21, MLK$^+$19, MLK$^+$19] also attempt to perform detection based on the representation. VeloFCN [LZX16] is the pioneering work to perform 3D objection using the range view, which transforms point cloud to the range image and then applies 2D convolution to detect 3D objects. After that, some following works [MLK$^+$19, FXW$^+$21] are proposed to narrow the performance gap between RV-based and BEV-based detectors. LaserNet [MLK$^+$19] models the distribution of 3D box corners to capture their uncertainty, resulting in more accurate defections. RCD [BSM$^+$21] introduces the range-conditioned dilation mechanism to dynamically adjust the dilation rate in terms of the measured range, which can alleviate the scale-sensitivity issue of the RV-based detectors. RangeDet [FXW$^+$21] further proposes the Range Conditioned Pyramid to mitigate the scale-variation issue and utilizes the Meta-Kernel convolution to better exploit the 3D geometric information of the points. To our knowledge, these existing RV-based detectors only take into consideration the single frame point cloud and neglect the substantial improvements brought by the multi-frame fusion as shown in BEV-based detectors.

# 3 Our Approach

## 3.1 Range View Representation

Given a LiDAR point $(x, y, z)$ in the Cartesian coordinate system with the $z$-axis pointing upward, it can be uniquely transformed to the spherical coordinates $(r, \theta, \phi)$ with

$$r = \sqrt{x^2 + y^2 + z^2}, \; \theta = \mathrm{atan2}(y, x), \; \phi = \mathrm{atan2}(z, \sqrt{x^2 + y^2}), \quad (1)$$

where $r$, $\theta$, and $\phi$ are the range, azimuthal angle, and inclination angle, respectively, as shown in Fig. 1(left). The LiDAR samples the points with a fixed number of beams (denoted by $m$), each of which has a fixed inclination angle. These LiDAR beams synchronously rotate around the $z$-axis uniformly to obtain a $360°$ horizontal field of view and the LiDAR measures a certain number of times (denoted by $n$) per scan cycle (*i.e.*, per frame). Thus, the difference of adjacent measurements' azimuthal angles is $360/n°$. For example, on the nuScenes dataset, the LiDAR measures $n = 1086$ times per scan cycle and has $m = 32$ beams. The inclination angles of these beams are evenly spaced from $-30.67°$ to $10.67°$, inclusive. Note that the inclination angles are not always evenly spaced and subject to the specifications of the LiDAR.

Given the regularity of the azimuthal and inclination angles for a given LiDAR, we can discretize the azimuthal angles with $n$ bins, and inclination angles with $m$ bins, respectively. Let $(i, j)$ be the indices of the bins for azimuthal and inclination angles, respectively. By computing all the pairs of $(i, j)$ of the points in a single scan cycle, we can fill these points into a 2D image $I \in \mathbb{R}^{m \times n \times C}$ (*i.e.*, the range image), where $C = 9$ consists of the original Cartesian coordinates $(x, y, z)$, the spherical coordinates $(r, \theta, \phi)$, the reflected intensity $i$, the existence $e$ of the point, and a relative timestamp $t$. The existence denotes whether or not the location is filled by a point, and the relative timestamp is only valid in the multi-frame point cloud inputs and denotes the time difference between the frame containing this point and the current frame. In practice, the vehicle itself is often in motion, and this causes that some points might be projected to the same bins on the range image. In this case, we keep the one with the minimal distance to the vehicle.

## 3.2 Multi-round Range View Projection (MRV)

The point cloud of a single frame is often sparse in the 3D space and of low resolutions. In order to improve the detection performance, the current frame is often combined with several previous frames as the network's inputs [YZK21, ZT18]. Taking the nuScenes dataset as an example, the methods on this dataset often take as the inputs 10 frames, which is composed of the current frame and previous 9 frames, including ~240K points in total. The crucial issue of the range view representation is the collision that multiple points fall into the same bin happens much more frequently in the multi-frame case. For instance, on nuScenes, only ~28K points are finally kept in the range image and ~90% of the points are discarded due to the collision. The decimation makes the multi-frame point cloud have almost the same number of valid points with the single-frame version, which is the dominant reason that RV-based methods cannot enjoy the benefit of multi-frame inputs.

In order to cope with this crucial issue, we propose the Multi-round Range View (MRV) projection mechanism. To be specific, we first project the points with the method in Sec. 3.1. Then, instead of discarding the rejected points, we project them again with the same process and put them in another group of nine channels. This projection process is repeated until a sufficient number of points are kept. On the nuScenes dataset, this is repeated five times and the percentage of the retained points can be improved from ~10% to more than 50%. In theory, as long as we continue the process, all points can be kept. However, we found that the performance is saturated after 5 repetitions on the nuScenes dataset. Finally, the resulting range images of the five rounds are concatenated along the channel dimension and used as the inputs. Additionally, one caveat is that the points of the current frame should have the highest priority wherever the collision happens because the points of previous frames are stale and might not reflect the current status of the world. Despite being a very simple treatment, it significantly affects the effectiveness of the multi-frame fusion as shown in our experiments.

## 3.3 Modality-wise Convolutions

As mentioned before, each pixel on the single-frame range image contains nine channels. Different the RGB image, whose three channels are of the same modality (*i.e.*, in the color space) and correlated

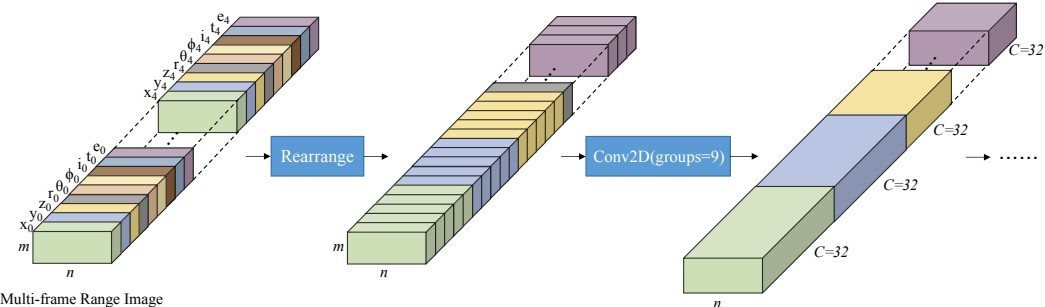

Multi-frame Range Image

Figure 2: **Modality-wise convolutions with a multi-frame range image.** As we can see, we first rearrange the channels of the multi-frame range image so that the channels with the same type from different frames (*e.g.*, $x_0$, $x_1$, ... $x_{T-1}$) are adjacent. Then, two successive 2D conv. layers with the number of groups being 9 (which is equal to the number of different channel types) are used to process these channels separately, mapping each channel type to a 32-channel features. Finally, the features of these channel types are merged by a $1 \times 1$ conv. layer.

for the manifold of natural images, the nine channels of the range image are not like that and belong to five modalities, *i.e.*, $[x, y, z]$, $[r, \theta, \phi]$, $[i]$, $[e]$ and $[t]$, respectively, where the channels in the one pair of square brackets are of the same modality. For example, the correlation between the reflected intensity $i$ and the coordinate $x$ does not make sense. Further, even the different channel types in the same modality are orthogonal and less correlated as well. For instance, it is difficult to say there is a relationship between the azimuthal angle $\theta$ and the range $r$ of a point. As a result, one channel type should be viewed as an individual "modality". By default, the conv. layer simultaneously computes spatial correlations and cross-channel correlations. However, as shown before, the channels of the range image are less correlated, and thus, it is not reasonable to use the default conv. layer here, and the channels of the range image should be processed separately. This can be easily implemented with the grouped convolutions. Here, we term it *modality-wise convolution* because it is based on the "modalities". Once the high-level semantic features of these modalities are individually obtained, we can aggregate the features of these modalities with a $1 \times 1$ conv. layer (*i.e.*, point-wise convolution) for further abstract analysis.

In this way, the modality-wise convolution is analogous to the widely-used depth-wise convolution, but we highlight that the underlying nature is distinct. Our modality-wise convolution instantiates a reasonable *inductive bias* based on the prior that different modalities are less relevant, and thus it is expected to simultaneously improve both the effectiveness and efficiency, as shown in our experiments. This also leads to the fact that the modality-wise convolution can only be placed at the beginning of the network, where the channels are interpretable and have an explicit modality. In contrast, the depth-wise convolution (together with the point-wise convolution) is often viewed as an efficient approximation of the full conv. layer and thus it does not usually yield improved performance and can appear anywhere. Lastly, when it comes to the multi-frame point clouds, the channels of the same type from different frames should be handled together because they are closely correlated. The whole procedure of the modality-wise convolution is illustrated in Fig. 2.

### 3.4 Overall Architecture

The overall architecture of FCOS-LiDAR is shown in Fig. 3. FCOS-LiDAR follows the spirit of the anchor-free detector FCOS [TSCH19] in the image-based object detection and is a standard fully convolutional network [LSD15].

Taking a (multi-frame) range image $I \in \mathbb{R}^{m \times n \times (T \times C)}$ as an example, where $T$ is the number of the used frames, we first forward the range image through our backbone network, termed LiDAR-Net. LiDAR-Net is adapted from ResNet-50 [HZRS16]. Specifically, before the first conv. layer of ResNet-50, we insert three branches of the modality-wise convolutions with three dilation rates being 1, 3, and 6, respectively. The branches with various dilation rates aim to capture the multi-scale context, whose outputs are summed up. Then, the first two $2\times$ downsamplings of ResNet-50 are removed, which is of great importance due to the low resolutions of the range images. Moreover, we change

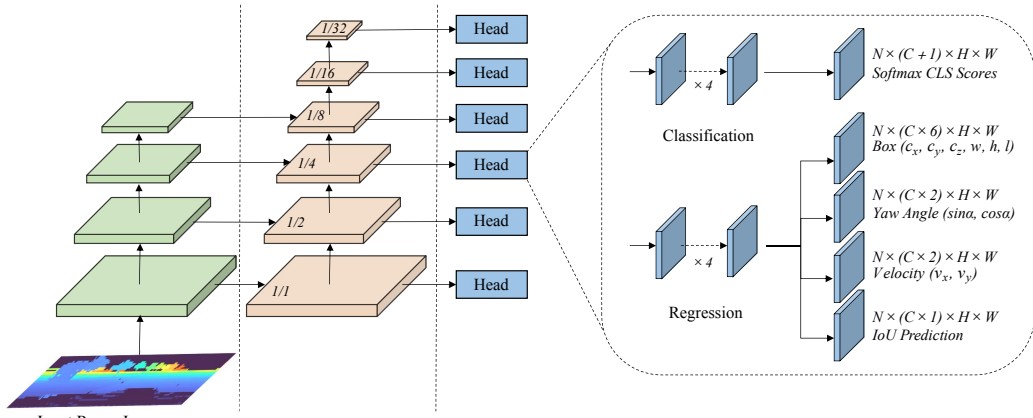

Figure 3: **Overall architecture.** The overall architecture of FCOS-LiDAR resembles the 2D image-based detector FCOS [TSCH19]. By taking as input an range image, the network obtains the multi-level FPN features, and then the classification and regression branches are attached to these feature levels to predict the final 3D boxes. Different from FCOS, the weights of the detection heads are not shared between the FPN levels as mentioned in Sec. 3.4. In addition, the class-specific regression heads are used instead of the class-agnostic ones in FCOS.

the numbers of blocks of ResNet-50's four stages from $(3, 4, 6, 3)$ to $(4, 4, 1, 1)$ and stop doubling the number of channels in the third and fourth stages because we found that the final performance is not sensitive to the capacity of the later stages but quite sensitive to that of the early stages. This ravels one of the important difference between the LiDAR-based and image-based object detection tasks. For object detection in RGB images, more convolutions in the later stages are often required to transform the RGB pixels into the highly semantic and abstract features that can be used to obtain the geometries of the objects. However, the LiDAR points themselves are already geometric points and thus that many convolutions in the later stages are no longer needed. Thus, we can instead allocate the capacity in the later stages to the early stages (before any downsampling) to better incorporate the geometric information carried by the raw points. Finally, the four levels of feature maps from LiDAR-Net's four stages are used, denoted by $C_2$, $C_3$, $C_4$, and $C_5$, respectively.

Next, following FCOS, the four levels of feature maps are sent into a feature pyramid network (FPN) [LDG$^+$17] to obtain six levels of pyramid feature maps denoted by $P_2$, $P_3$, $P_4$, $P_5$, $P_6$, and $P_7$. Their spatial downsampling ratios to the input are $^1/_1$, $^1/_2$, $^1/_4$, $^1/_8$, $^1/_{16}$, and $^1/_{32}$, respectively. Then, similar to FCOS, the classification and regression heads, each with four $3 \times 3$ conv. layers of channel 64 and a final prediction conv. layer (for the classification or regression), are attached to these feature levels. Unlike the heads in image-based detectors, whose weights are shared between these feature levels, it is important in the LiDAR-based detector to untie these weights. This is another important difference between image-based and LiDAR-based detectors. In image-based detectors, different sizes of objects can be normalized to similar sizes of objects by the downsampling the image with corresponding factors. That is what the "pyramid" means in the FPN. Thus, the image-based detectors can share the weights of the detection heads because the objects have been normalized to similar sizes after the FPN. However, in LiDAR-based detection, the objects' sizes cannot be normalized in this way because the sizes of the objects are determined by the 3D points' coordinate values in them and downsampling the range images cannot alter their real sizes in the 3D space. Therefore, it is no longer reasonable to share the detection heads between these FPN levels. This is also confirmed in our experiments.

**Training Targets Computation.** Similar to FCOS [TSCH19], we need to assign the training targets to each location of the feature maps computed with the range image. The training targets computation is described using the nuScenes dataset as an example. On nuScenes, an object's ground-truth 3D box is parameterized by $(c_x^*, c_y^*, c_z^*, w^*, h^*, l^*, \alpha^*)$, where $(c_x^*, c_y^*, c_z^*)$ is the 3D center of the box, and $w, h, l$, and $\alpha$ respectively are the width, height, length, and the yaw angle around the $z$ axis. First, each location on the feature maps is mapped to the pixel location on the range image by multiplying them by the feature maps' downsampling ratio. Next, we use the 3D LiDAR point projected by

the *first-round* MRV projection as the 3D point of the pixel. If the pixel's 3D point is contained in an object's 3D box, the feature location is responsible for the object and predicts its category, 3D box and etc.. Here, the 3D box regression targets of the feature location are relative to the 3D point coordinates of the pixel and are defined as

$$\Delta c_x = c_x^* - x_0, \ \Delta c_y = c_y^* - y_0, \ \Delta c_z = c_z^* - z_0, \ t_w = \log(w^*), \ t_h = \log(h^*), \ t_l = \log(l^*), \ (2)$$

where $(x_0, y_0, z_0)$ are the 3D point coordinates of the range image pixel. Similarly, the regression targets of the yaw angle $\alpha$ are also relative to the azimuthal angle of the pixel, and following the convention [YZK21], we decouple the relative azimuthal angle into $(\sin \Delta\alpha, \cos \Delta\alpha)$. On nuScenes, we also need to predict the velocity vector $(v_x^*, v_y^*)$ of the object, which are used as the training targets as is. Other locations on the feature maps that correspond to 3D points not in any 3D box are used as the negative samples. Note that all the pixels within a object's 3D box actually form a 2D mask on the range image.

**Network Outputs.** The nuScenes dataset has $C = 10$ classes of interest, which is predicted by the classification head followed by a $\mathrm{softmax}$ layer (the upper head in Fig. 3). The other regression targets are predicted by four sibling output heads of the regression branch, respectively, as shown in Fig. 3. Here, we use the class-specific regression predictions and thus the number of the output channels is amplified by $C$ times. Moreover, following [GLW$^+$21, RDGF16], each positive pixel also predicts the intersection-over-union (IoU) between the predicted 3D box and ground-truth one, which is multiplied to the classification score before the non-maximum suppression (NMS).

**Loss Functions.** The classification predictions are supervised with the cross entropy (CE) loss. Following [GLL$^+$21], we dynamically reassign the classification labels during training. Specifically, for a ground-truth 3D box, only the top $K$ pixels whose predicted 3D boxes have the lowest costs with the ground-truth box are assigned with the positive labels and other pixels are considered negative, where the cost is defined as the summation of the classification loss and the opposite of the IoU between the predicted boxes and the ground-truth box, and $K$ is dynamically calculated, being the summation (rounded to an integer) of the highest $Q = 20$ IoUs between the predicted boxes and the ground-truth box. For the 3D box regression, we make use of both the IoU loss [YJW$^+$16] and $L_1$ loss. Following [TSCH19], the IoU predictions are penalized with the binary cross entropy (BCE) loss since they are in the range $[0, 1]$.

**Inference.** The inference is very similar to that of the 2D image-based detector FCOS. To be specific, the range images are forwarded through the network and the aforementioned predictions are obtained. The predictions are filtered in terms of the classification scores and only the predictions with the score greater than 0.01 are kept. The 3D boxes are restored by inverting the computation of the training targets. Then, the NMS on the 3D boxes is applied with threshold 0.2 to remove the duplications. Finally, the top 500 predictions with the highest scores are used as the final predictions.

## 4  Experiments

We conduct experiments on the nuScenes dataset [CBL$^+$20], which contains 1000 scenes with 700, 150, and 150 scenes for training, validation, and testing, respectively. The training, validation, and testing sets have 28K, 6K, 6K keyframes annotated with 10 classes, respectively. For all ablation experiments, we train the models on the training set and report the performance on the validation set unless specified. The metrics of the 3D detection task are mean Average Precision (mAP) and the nuScenes detection score (NDS). The LiDAR sensor of the nuScenes dataset has $m = 32$ beams and $n = 1086$ measurements per scan cycle. The mAP is based the center distances on the bird-eye view at thresholds 0.5m, 1m, 2m, 4m in place of the box IoUs. NDS is a weighted average of mAP, the translation error, the scale error, the orientation error, the velocity error, and the box attributes error.

**Implementation Details.** Unless specified, following [YZK21], FCOS-LiDAR is trained by 40 epochs with the AdamW [LH17] optimizer under the MMDetection3D framework [Con20], which takes ∼26 hours on 8 A100 GPUs. The one-cycle learning rate policy [ST19] with initial learning rate $10^{-3}$ is used. The learning rate gradually increases to 0.01 in the first 40% epochs and then gradually decreases to $10^{-7}$ in the rest of the training process. The weight decay is 0.01, and the momentum ranges from 0.85 to 0.95. In addition, due to the low vertical resolution of the range image on the nuScenes dataset, we upscale the range image in the vertical direction by 2 with the nearest interpolation. During training, the point cloud is randomly flipped along both the $x$ and $y$

Table 1: **Multi-round range view (MRV) projection.** Time: the elapsed time of MRV.

| #Rounds | Time (ms) | Single-frame | | Multi-frame | |
|---|---|---|---|---|---|
| | | mAP (%) | NDS (%) | mAP (%) | NDS (%) |
| 1 | **0.66** | 53.14 | 51.52 | 55.02 | 61.27 |
| 3 | 0.72 | **53.76** | **53.62** | 56.01 | 62.82 |
| 5 | 0.76 | 53.42 | 53.40 | **57.08** | 63.15 |
| 7 | 0.76 | 53.06 | 53.10 | 56.99 | **63.47** |

Table 2: **Whether to make the points of the current frame have the highest priority.** The per-category mAP results are reported. "C.V.", "Ped." and "C.T." indicate "construction vehicle", "pedestrian" and "traffic cone", respectively.

| Prioritized? | mAP(%) | NDS(%) | Car | Truck | Bus | Trailer | C.V. | Ped. | Motor | Bicycle | T.C. | Barrier |
|---|---|---|---|---|---|---|---|---|---|---|---|---|
| | 54.29 | 62.31 | 76.8 | 47.3 | 62.0 | 32.0 | 16.5 | 80.3 | 53.8 | **38.2** | 70.1 | 65.9 |
| ✓ | **57.08** | **63.15** | **82.1** | **52.3** | **65.2** | **33.6** | **18.3** | **84.1** | **58.5** | 35.3 | **73.4** | **67.9** |

axes and rotated in the range $[-\pi, \pi]$, as well as globally scaled by a random factor from $[0.95, 1.05]$. The ground-truth copy-paste data augmentation from [YML18] is also used. For multi-frame point cloud, we use 10 sweeps in total, as in previous works [YZK21, LVC+19]. The inference time is measured on a 3090Ti GPU with batch size 1.

## 4.1 Multi-round Range View Projection

Here, we conduct experiments to demonstrate the effectiveness of the proposed multi-round range view (MRV) projection. As shown in Table 1, in the single-frame settings, MRV is also helpful, for example, by increasing the number of rounds from 1 to 3, the mAP can be boosted by 0.62%. This is due to the fact that the vehicle is often in motion and the aforementioned collisions happen within a single frame as well. As you can see, if one round is used, the multi-frame fusion can improve the mAP by a substantial margin ∼1.9% (from 53.14% to 55.02%). The improvement can be dramatically increased to 3.9% mAP if we use 5 rounds in MRV (57.8% mAP), due to the fact that 5-round MRV can keep much more points in the multi-frame point clouds as mentioned in Sec. 3.2. This confirmed the effectiveness of MRV. Note that elapsed time of MRV is insensitive to the number of rounds as shown in Table 1.

More importantly, in the RV-based multi-frame settings, it is crucial to assign the highest priority to the points from the current frame when the collision happens. As shown in Table 2, without this treatment, the performance of the multi-frame fusion is significantly dropped from 57.08% to 54.29% in mAP. Note that the single-frame counterpart can already achieve mAP 53.42%. We argue that the neglect of this point in previous works is one of the main reasons that RV-based detectors can barely enjoy the benefit of multi-frame fusion.

## 4.2 Modality-wise Convolutions

Table 3: **Modality-wise convolutions.** "Multi-round": whether to group together the channels with the same type from multiple projection rounds. Note that all the items here are with the multi-frame fusion and multi-round projections. The only difference is the way to group the channels. Time: the latency of this module.

| Modality groups | Multi-round | Time (ms) | mAP (%) | NDS (%) |
|---|---|---|---|---|
| $[x, y, z, r, \theta, \phi, i, t, e]$ | ✓ | 6.0 | 56.35 | 62.65 |
| $[x, y, z], [r, \theta, \phi], [i], [t], [e]$ | ✓ | 4.2 | 56.34 | 63.07 |
| $[x], [y], [z], [r], [\theta], [\phi], [i], [t], [e]$ | ✓ | **3.2** | **57.08** | **63.15** |
| $[x], [y], [z], [r], [\theta], [\phi], [i], [t], [e]$ | | 3.8 | 56.83 | **63.15** |

The experimental results of our modality-wise convolutions are shown in Table 3. If we use the full convolutions here, which compute the correlation between all the channels of the range image, FCOS-LiDAR can achieve 56.35% in mAP. By splitting the channels into various modalities (2nd row), the performance can be slightly improved, *i.e.*, from NDS 62.65% to 63.07%. Moreover, due to

Table 4: **Varying the number of the conv. layers in the modality-wise convolutions.** Time: the latency of the modality-wise convolutions.

| #Conv. | Time (ms) | mAP(%) | NDS(%) | Car | Truck | Bus | Trailer | C.V. | Ped. | Motor | Bicycle | T.C. | Barrier |
|---|---|---|---|---|---|---|---|---|---|---|---|---|---|
| 1 | **2.0** | 55.81 | 61.98 | 81.8 | 51.4 | 65.2 | 30.7 | 16.9 | 83.4 | 56.0 | 32.8 | 71.9 | 68.1 |
| 2 | 3.2 | **57.08** | 63.15 | 82.1 | **52.3** | 65.2 | **33.6** | 18.3 | **84.1** | **58.5** | **35.3** | **73.4** | 67.9 |
| 3 | 4.3 | 56.71 | **63.25** | **82.6** | 51.4 | **65.3** | 31.9 | **18.5** | 83.9 | 57.3 | 34.9 | 72.9 | **68.4** |

Table 5: **Multi-scale context aggregation.**

| Dilations | mAP(%) | NDS(%) | Car | Truck | Bus | Trailer | C.V. | Ped. | Motor | Bicycle | T.C. | Barrier |
|---|---|---|---|---|---|---|---|---|---|---|---|---|
| $(1,)$ | 55.87 | 62.68 | 81.8 | 51.0 | 64.1 | 32.6 | 16.9 | 83.8 | 56.3 | 31.2 | 72.7 | **68.3** |
| $(1,3)$ | 56.39 | 62.96 | **82.3** | 52.1 | **65.3** | 32.8 | 17.1 | 83.7 | 58.2 | 32.5 | 72.0 | 67.8 |
| $(1,3,6)$ | **57.08** | **63.15** | 82.1 | **52.3** | 65.2 | **33.6** | **18.3** | **84.1** | 58.5 | **35.3** | **73.4** | 67.9 |
| $(1,1,1)$ | 56.63 | 63.12 | 82.1 | 51.5 | 65.2 | 32.9 | 17.1 | 83.7 | **58.8** | 33.7 | 73.1 | 68.0 |

the fact that even the different channel types of the same modality are orthogonal and less correlated, we process each channel type individually. As shown in the table, the performance can be further boosted from mAP 56.34% to 57.08% (3rd row) while the lowest latency is achieved. Finally, the last row shows the results if we do not consider the channels with the same type from multiple rounds together, where the mAP is slightly worse. In Table 4, we vary the number of the conv. layers in the modality-wise convolutions. As we can see, using two conv. layers achieves the best performance here.

In addition, as mentioned before, we employ multiple modality-wise convolution branches with various dilation rates in parallel to capture the multi-scale context. The experiments are shown in Table 5. As we can see, compared with the one using only one dilation rate, using multiple dilation rates can improve the performance by more than 1% in mAP (from 55.87% to 57.08%).

## 4.3 Untied Weights of Detection Heads

Table 6: **Whether to untie the weights of the detection heads.**

| Untied? | mAP(%) | NDS(%) | Car | Truck | Bus | Trailer | C.V. | Ped. | Motor | Bicycle | T.C. | Barrier |
|---|---|---|---|---|---|---|---|---|---|---|---|---|
| | 56.44 | 63.09 | 82.0 | 51.1 | 64.3 | 31.1 | **18.3** | 83.8 | 57.9 | 34.7 | 72.9 | **68.4** |
| ✓ | **57.08** | **63.15** | **82.1** | **52.3** | **65.2** | **33.6** | 18.3 | **84.1** | **58.5** | **35.3** | **73.4** | 67.9 |

As mentioned before, it is better to untie the weights of the detection heads between the FPN levels in the LiDAR-based detector. This is confirmed in Table 6. As we can see, by untying the weights, the performance can be improved from mAP 56.44% to 57.08%. This ravels one of the interesting differences between image-based and LiDAR-based detectors because the shared detection heads between FPN levels often achieve better performance in image-based detectors [TSCH19, LDG+17].

Table 7: **Inference time breakdowns.** We compare against the state-of-the-art BEV-based Center-Point [YZK21], which is trained with exactly the same strategies. We use the CenterPoint implementation in MMDetection3D [Con20] with sparse convolution version 1.2.1 [SPC21]. FCOS-LiDAR is faster as well as competitive in the multi-frame setting (and superior in the single-frame setting).

| Method | Voxel/MRV(ms) | Backbone(ms) | Heads(ms) | Overall(ms) | mAP(%) | NDS(%) |
|---|---|---|---|---|---|---|
| *single-frame point cloud* | | | | | | |
| CenterPoint [YZK21] | 1.62 | 41 | 8 | 50.62 | 52.83 | **53.86** |
| FCOS-LiDAR (**Ours**) | **0.63** | **31** | **7** | **38.63** | **53.42** | 53.40 |
| *multi-frame point cloud* | | | | | | |
| CenterPoint [YZK21] | 1.95 | 63 | 9 | 73.95 | **60.40** | **67.25** |
| FCOS-LiDAR (**Ours**) | **0.76** | **31** | **7** | **38.76** | 57.08 | 63.15 |

Table 8: **Comparisons with state-of-the-art methods on the nuScenes test set.** The results are directly quoted from their original papers. All other methods on nuScenes rely on BEV or multiple views because previous RV-only methods are unable to handle the multi-frame fusion on this dataset. As you can see, our RV-based FCOS-LiDAR can achieve competitive performance with state-of-the-art BEV-based detectors.

| Method | mAP(%) | NDS(%) | Car | Truck | Bus | Trailer | C.V. | Ped. | Motor | Bicycle | T.C. | Barrier |
|---|---|---|---|---|---|---|---|---|---|---|---|---|
| PointPillars [LVC+19] | 30.5 | 45.3 | 68.4 | 23.0 | 28.2 | 23.4 | 4.1 | 59.7 | 27.4 | 1.1 | 30.8 | 38.9 |
| SSN [ZMW+20] | 46.3 | 56.9 | 80.7 | 37.5 | 39.9 | 43.9 | 14.6 | 72.3 | 43.7 | 20.1 | 54.2 | 56.3 |
| CVCNet [CSCY20] | 55.3 | 64.4 | 82.7 | 46.1 | 46.6 | 49.4 | 22.6 | 79.8 | 59.1 | 31.4 | 65.6 | 69.6 |
| CBGS [YZK21] | 52.8 | 63.3 | 81.1 | 48.5 | 54.9 | 42.9 | 10.5 | 80.1 | 51.5 | 22.3 | 70.9 | 65.7 |
| CenterPoint [YZK21] | 58.0 | 65.5 | 84.6 | 51.0 | 60.2 | 53.2 | 17.5 | 83.4 | 53.7 | 23.7 | 76.7 | 70.9 |
| AFDetV2 [HDG+22] | 62.4 | 68.5 | **86.3** | 54.2 | 62.5 | 58.9 | 26.7 | **85.8** | 63.8 | 34.3 | 80.1 | 71.0 |
| S2M2-SSD [ZHJF22] | 62.9 | 69.3 | **86.3** | 56.0 | **65.4** | 59.8 | 26.2 | 84.6 | 61.6 | 36.4 | 77.7 | 75.1 |
| VISTA [DLSJ22] | 63.0 | 69.8 | 84.4 | 55.1 | 25.1 | 63.7 | 54.2 | 71.4 | **70.0** | **45.4** | 82.8 | 78.5 |
| TransFusion [XBT22] | **65.5** | **70.2** | 86.2 | **56.7** | 28.2 | **66.3** | **58.8** | 78.2 | 68.3 | 44.2 | **86.1** | **82.0** |
| FCOS-LiDAR (c128) | 60.2 | 65.7 | 82.2 | 47.7 | 52.9 | 48.8 | 28.8 | 84.5 | 68.0 | 39.0 | 79.2 | 70.7 |

## 4.4 Inference Time Comparisons

We compare the inference time of FCOS-LiDAR and the state-of-the-art BEV-based detector Center-Point [YZK21]. As shown in Table 7, the proposed MRV is faster than the voxelization in CenterPoint. It is worth noting that the voxelization algorithm reported here is *nondeterministic*, which cannot yield stable results but being significantly faster. In the official code of MMDetection3D [Con20], the deterministic voxelization takes ∼100ms for the multi-frame point clouds. In sharp contrast, the proposed MRV is always *deterministic* and highly efficient. Moreover, due to the compactness of the range image, our network can be implemented with the standard convolutions alone, thus being much more efficient than CenterPoint using the sparse convolutions as shown in the table. The elapsed time of the post-processing is omitted here as it is closely similar in both methods.

## 4.5 Comparisons with State-of-the-art Methods

We further compare FCOS-LiDAR with other state-of-the-art methods on the nuScenes test set. For the model on the test set, we increase the number of channels in the detection heads from 64 to 128, which improve the validation mAP by ∼0.6% with slightly longer latency. Additionally, we remove the copy-paste data augmentation in the last 5 epochs during training as in [WMZY21] (termed as the "fade strategy" in [WMZY21]). This can improve the performance by about 2% mAP. As shown in Table 8, FCOS-LiDAR achieves competitive performance with other state-of-the-art BEV-based methods. Note that in order to leverage the multi-frame fusion on nuScenes, all other methods on nuScenes rely on BEV (or the multi-view fusion). FCOS-LiDAR is the first RV-only method that is able to benefit from the multi-frame fusion.

# 5 Conclusion

We have presented an efficient range-view-based 3D object detector FCOS-LiDAR. FCOS-LiDAR shows the challenging LiDAR-based object detection can also be solved with the standard convolutions alone, similar to what we have done in the image-based 2D object detection. We also for the first time show the RV-based 3D detector can also enjoy the benefit of the multi-frame fusion with the proposed MRV. We hope our strong results can encourage the community to pay more attention to this promising direction.

**Societal Impacts.** This paper presents a method that can locate the 3D location of the objects of interest in LiDAR point clouds. This technique might be abused for military purposes, for example, on lethal autonomous weapons.

**Acknowledgments.** C. Shen's participation was in part supported by a major grant from Zhejiang Provincial Government.

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
