# Supplementary Materials for
# Fully Convolutional One-Stage 3D Object Detection on LiDAR Range Images

## 1 Errata

We apologize that we mistook the model used to obtain the results on the nuScenes test set. A flawed model was used and thus the results in the submitted paper are not of the real final model mentioned in the paper. The correct results corresponding to the real final model are much better and are updated in Table 1. This issue has no effect on other models and all other results are correct.

Table 1: **Comparisons with state-of-the-art methods on the nuScenes test set.** The results are directly quoted from their original papers.

| Method | mAP(%) | NDS(%) | Car | Truck | Bus | Trailer | C.V. | Ped. | Motor | Bicycle | T.C. | Barrier |
|---|---|---|---|---|---|---|---|---|---|---|---|---|
| PointPillars [1] | 30.5 | 45.3 | 68.4 | 23.0 | 28.2 | 23.4 | 4.1 | 59.7 | 27.4 | 1.1 | 30.8 | 38.9 |
| SSN [2] | 46.3 | 56.9 | 80.7 | 37.5 | 39.9 | 43.9 | 14.6 | 72.3 | 43.7 | 20.1 | 54.2 | 56.3 |
| CVCNet [3] | 55.3 | 64.4 | 82.7 | 46.1 | 46.6 | 49.4 | 22.6 | 79.8 | 59.1 | 31.4 | 65.6 | 69.6 |
| CBGS [4] | 52.8 | 63.3 | 81.1 | 48.5 | 54.9 | 42.9 | 10.5 | 80.1 | 51.5 | 22.3 | 70.9 | 65.7 |
| CenterPoint [4] | 58.0 | 65.5 | **84.6** | **51.0** | **60.2** | **53.2** | 17.5 | 83.4 | 53.7 | 23.7 | 76.7 | **70.9** |
| ~~FCOS-LiDAR (c128)~~ | ~~58.8~~ | ~~64.8~~ | ~~81.7~~ | ~~45.8~~ | ~~52.3~~ | ~~49.0~~ | ~~27.5~~ | ~~83.7~~ | ~~64.1~~ | ~~35.8~~ | ~~77.9~~ | ~~70.0~~ |
| FCOS-LiDAR (c128) | **60.2** | **65.7** | 82.2 | 47.7 | 52.9 | 48.8 | **28.8** | **84.5** | **68.0** | **39.0** | **79.2** | 70.7 |

## 2 More Experiments

As mentioned before, for the model on the test set, we use 128 channels in the detection heads, which improve the performance from 57.08% to 57.71% mAP with 6ms more latency, as shown in Table 2. In Table 3, we vary the number of the conv. layers in the modality-wise convolutions. As we can see, using two conv. layers achieves the best performance here.

In addition, we compare with more popular 3D detectors by keeping the training schedule the same. As shown in Table 4, FCOS-LiDAR significantly outperforms these methods.

Table 2: **Varying the number of the channels in the detection heads.** Time: the latency of the detection heads.

| #Channels | Time (ms) | mAP(%) | NDS(%) | Car | Truck | Bus | Trailer | C.V. | Ped. | Motor | Bicycle | T.C. | Barrier |
|---|---|---|---|---|---|---|---|---|---|---|---|---|---|
| 64 | **7** | **57.08** | 63.15 | 82.1 | 52.3 | 65.2 | 33.6 | **18.3** | 84.1 | 58.5 | **35.3** | 73.4 | 67.9 |
| 128 | 13 | 57.71 | **64.09** | **82.9** | **53.4** | **66.5** | **34.7** | 18.1 | 84.1 | 58.5 | 35.2 | **74.3** | **69.4** |

Table 3: **Varying the number of the conv. layers in the modality-wise convolutions.** Time: the latency of the modality-wise convolutions.

| #Conv. | Time (ms) | mAP(%) | NDS(%) | Car | Truck | Bus | Trailer | C.V. | Ped. | Motor | Bicycle | T.C. | Barrier |
|---|---|---|---|---|---|---|---|---|---|---|---|---|---|
| 2 | **3.2** | **57.08** | 63.15 | 82.1 | **52.3** | 65.2 | **33.6** | 18.3 | **84.1** | **58.5** | **35.3** | **73.4** | 67.9 |
| 3 | 4.3 | 56.71 | **63.25** | **82.6** | 51.4 | **65.3** | 31.9 | **18.5** | 83.9 | 57.3 | 34.9 | 72.9 | **68.4** |

Table 4: **Comparisons with more popular methods on the nuScenes validation set.** To make fair comparisons, the other methods are also trained for 40 epochs using the implementation in MMDetection3D.

| Method | mAP(%) | NDS(%) | Car | Truck | Bus | Trailer | C.V. | Ped. | Motor | Bicycle | T.C. | Barrier |
|---|---|---|---|---|---|---|---|---|---|---|---|---|
| PointPillars [1] | 41.3 | 54.9 | 81.3 | 37.7 | 48.5 | 29.9 | 8.5 | 72.7 | 38.9 | 30.6 | 33.5 | 37.2 |
| SSN [2] | 45.0 | 57.3 | **82.3** | 44.9 | 59.8 | 29.6 | 12.8 | 69.8 | 47.9 | 23.2 | 24.8 | 40.3 |
| FCOS-LiDAR | **57.08** | **63.15** | 82.1 | **52.3** | **65.2** | **33.6** | **18.3** | **84.1** | **58.5** | **35.3** | **73.4** | **67.9** |

## 3 Visualization

Some visualization results are shown in Fig. 1. As we can see, FCOS-LiDAR can work reliably under a wide variety of challenging circumstances.

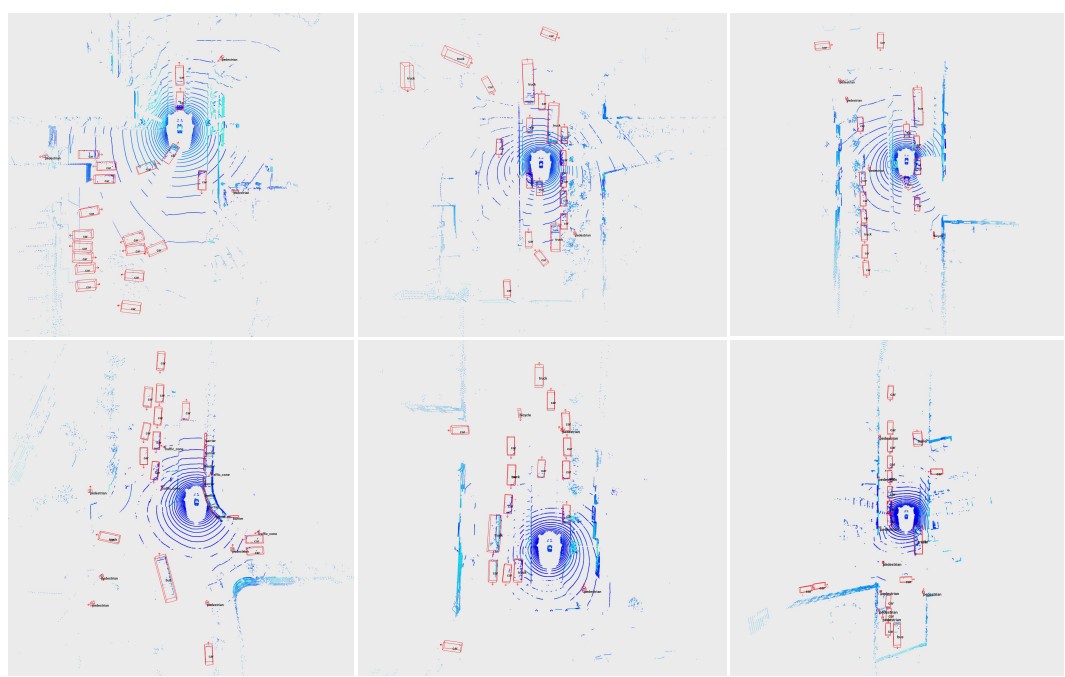

Figure 1: **Visualization results of FCOS-LiDAR on the nuScenes val set.**