# OpenReview forum: "Fully Convolutional One-Stage 3D Object Detection on LiDAR Range Images"
_NeurIPS.cc/2022/Conference — NeurIPS 2022 Accept_

### Official Review · Reviewer_pS8M · 2022-07-10

**Rating:** 7
**Confidence:** 4
**Soundness:** 4 excellent
**Presentation:** 4 excellent
**Contribution:** 3 good

**Summary:**

Because 3D detectors for autonomous driving need to perform quickly and have good detection performance in a limited computational environment, Voxel-based and BEV-based neural detectors have been suggested in existing works. However, the authors proposed a novel type of 3D detector that can maximize the utility of range view inputs. They additionally propose a multi-round range view projection to contain more points within a single range view image and the corresponding stem layer which is called modality-wise convolution.

**Questions:**

**How to find the points corresponding to a given range view pixel?** I reckon that the method of finding a point corresponding to a pixel in a range view image with various scales is more complicated, unlike BEV. For feature maps that are smaller than the original size, there may be various options for how to determine the depth for a pixel. This is because points with different depths can be assigned simultaneously within one pixel in this case. In other words, it can introduce ambiguity when dealing with training targets. How you find the points that map to a pixel also affects dynamic allocation. Please provide a detailed explanation on this.

Are the authors willing to release the code for active research on range views?


**Limitations:**

They are aware of the dangerous uses of object detection. In addition, it would be good if future works parts were added for this field.

**Strengths And Weaknesses:**

Their view of LiDAR points will have implications for other researchers as well, and the proposed method is sufficient to serve as a baseline for range-view-based 3D detectors in terms of performance.

The proposed module and its motivation are well aligned. In particular, the backbone network for range view was reconstructed based on resnet50, and the explanation of its motivation is very interesting.

The authors designed a range view-based neural network through multi-round range view projection (MRV) that minimizes the point loss that may occur in the process of converting to range view images, and a specified 2D CNN network for lidar points.

However, when generating range-view images, a single pixel can contain multiple points as a range-view image is generated through multiple projections. In this regard, it is necessary to explain the target assignment process more clearly. For example, when two objects are assigned to one pixel, the authors need to describe how they are handled.

---

> ### Author Response · Authors · 2022-08-02
> **Responses to Reviewer pS8M**
>
> **Q1. How to handle the pixels each having multiple points?**
>
> Thank you for your question. You have a precise understanding of our work!
>
> Yes, a single pixel on the range image can have multiple 3D points due to the multi-round projection, which might belong to different objects. In this work, we only use the points in the *first-round projection* to compute the target assignments. If a pixel is assigned to multiple ground-truth boxes (this case is very rare in 3D object detection), the one with the minimum range view projection area is chosen. We did attempt to use the points from all rounds. However, both achieved a similar performance, as shown in the following table. The similar performance might be due to the fact that using points in the first-round projection is enough to compute target assignments and recall the target objects.
>
> |                  | mAP(%)         | NDS(%)         | Car           | Truck         | Bus           | Trailer       | C.V.          | Ped.          | Motor         | Bicycle       | T.C.          | Barrier       |
> |------------------|----------------|----------------|---------------|---------------|---------------|---------------|---------------|---------------|---------------|---------------|---------------|---------------|
> | all rounds       | 58.18          | 64.27          | 83.4          | 53.4          | 67.2          | 33.3          | 19.6          | 83.7          | 60.6          | **37.8** | **73.0** | **69.7** |
> | first round only | **58.49** | **64.64** | **83.5** | **54.6** | **67.4** | **34.9** | **19.7** | **84.2** | **62.2** | 36.3          | 72.4          | **69.7** |
>
> **Q2. How to find the points corresponding to a given range view pixel?**
>
> Although each location $(x, y)$ on the FPN feature maps with downsampling ratio $s > 1$ can be mapped to multiple pixels in the range image with the original size, we prescribe it is mapped to one exact pixel $(xs, ys)$ on the range image. One pixel on the original range image corresponds to one 3D point (only the points in the first-round projection are considered). This 3D point is used to compute the targets assignments. Thus, when we deal with training targets, there is not the ambiguity that one pixel simultaneously corresponds to multiple points with different depths. Then, we obtain the corresponding 3D point at this location and check whether the 3D point is inside any ground-truth box. If any, the ground-truth box is assigned as the target of this pixel. Otherwise, the pixel is background. If a point falls in multiple ground-truth boxes, as mentioned before, the one with the minimum area is chosen.
>
> **Q2. Open-source the code and future works.**
>
> Sure, we will definitely open-source our full code for active research on range views. We hope more researchers can pay attention to this direction. We will discuss the future works in the revision as well.
>
> Thank you for your time in reviewing our work! If you have more questions, please let us know.

---

> > ### Author Response · Authors · 2022-08-07
> > **Do you have further questions?**
> >
> > If you have more questions, please let us know. Thank you for your time!

---

> > ### Comment · Reviewer_pS8M · 2022-08-09
> > **Responses to Authors**
> >
> > Thank you for your detailed response and their decision to release the code. Despite the concerns below, I will keep the current score.
> >
> > Assuming we get the downsampled feature map through max pooling, the one-to-one mapping between points and features is guaranteed as the author mentioned. However, if average pooling or convolution with stride 2 is used, I still think that multiple points can be mapped. Can you provide a further explanation?
> >
> > p.s. Personally, I would like the current discussion and experiment related to the target assignment to be added to the manuscript or supplementary.

---

> > > ### Author Response · Authors · 2022-08-09
> > > **Responses to Reviewer pS8M**
> > >
> > > Thank you very much for your feedback. We will add the current discussion to our manuscript.
> > >
> > > In fact, the convolution with stride 2 can also keep the one-to-one mapping by assuming that each feature location is mapped to the center of the convolution's receptive field. For example, let us assume the kernel size is $3\times3$ and padding is $1$. The top-left location $(0, 0)$ on the feature maps is mapped to the top-left location of the input, i.e., $(0, 0)$; and the second point of the first row, i.e., $(0, 1)$, is mapped to $(0, 2)$ on the input. If you have more questions, feel free to post them here. Thank you very much!

---

> > > > ### Comment · Reviewer_pS8M · 2022-08-09
> > > > **Responses to Authors**
> > > >
> > > > Thank you for your prompt response!! I fully understand! Good luck!

---

### Official Review · Reviewer_Uzj9 · 2022-07-11

**Rating:** 4
**Confidence:** 3
**Soundness:** 2 fair
**Presentation:** 3 good
**Contribution:** 2 fair

**Summary:**

In this paper, the authors propose a range-view CNN based 3D detector. Instead of using a bird-eye-view representation (like most of current models), the sensor data are encoded into a range view. It allows a simple multi-round range view to consider a temporal information. The network applies modality-wise convolution, a kind of depth-wise convolution adapted to the proposed encoding. Experiment shows that the simple proposed model performs well

**Questions:**

Regarding the ablation study of table 3:  why do we have a higher time for Multi-Frame? Can you give more details about this? Do you use a sliding temporal window to compute one output for each time or do you compute some batches?

**Ethics Review Area:**

["I don’t know"]

**Limitations:**

/

**Strengths And Weaknesses:**

Strengths:
Even if other range-view model already exists, the contribution of the paper is the combination of this kind of representation with an encoding of the data combining cartesian + spherical coordinates + a multi-round range model that uses modality-wise convolution. Modality-wise convolution is a nice idea in order to reduce the size of the model (like with depth wise) but using an inductive bias assuming that channels encoding different information can be split in the convolution.

Weaknesses:
Regarding the ablation study of table 3, it seems that the best combination for modality groups is multi-frame on each channel. However, the mAP is slightly higher (less than 1%).

There is another issue with the experimental part.  The baseline models you compare with should be increased with the one given on the nuScene benchmark: https://www.nuscenes.org/object-detection?externalData=no&mapData=no&modalities=Any

Moreover, It should be better to give the performances of the models reported on this page instead of giving the one reported in the original papers to have updated criteria. If you do that, you will see that your model is not SOTA. However, it remains simple and this is interesting in some applications.

---

> ### Author Response · Authors · 2022-08-02
> **Responses to Reviewer Uzj9**
>
> **Q1. The ablation study of Table 3.**
>
> Sorry, we really appreciate it if you could make your questions clearer. We clarify that all the items in Table 3 are with the multi-frame fusion, and the difference is the way to group the channels. "Multi-frame" means that the channels from multiple range view projections but with the same type are grouped together. For example, $x_0$, $x_1$, and $x_2$ are put in one group because they are all the x-coordinate of a point, where $0$, $1$, or $2$ is the index of the range view projection.
>
> In addition, as noted in L287, all the inference time is measured with batch size 1 on a 3090Ti GPU. We do not use a sliding temporal window, and we only simply forward the range view images through the network. Additionally, the time differences in Table 3 (except the 1st row) are very small ($\leq$1ms), which might be dominated by the underlying implementations instead of the network designs. Thus, we hope you can pay more attention to the model's performance here, and what we choose is the model with the best mAP and NDS.
>
> **Q2. Comparisons with the nuScenes leaderboard results.**
>
> Thank you for your suggestions. We will add the leaderboard results to our paper.
>
> We would like to note that, when comparing our model with the baseline model, we strictly controlled the settings and make them consistent. The performance of the baseline used by us is actually much better than what others reported. We use the CenterPoint implementation from the official MMDetection3D repository (https://github.com/open-mmlab/mmdetection3d/tree/master/configs/centerpoint), whose performance is similar to the one in the original paper. We further align the training settings with ours, which yields better performance (improved from mAP 57.63% to 60.40% on the val set) as shown in Table 6.
>
> In addition, we want to note that the methods on the leaderboard often make use of model ensembles, test-time augmentation and other tricks (e.g., PointPainting) to attain good performance. Our method has no bells and whistles and can be easily deployed in practice. Also, the results on the leaderboard are not peer-reviewed.
>
> Hope our responses can address your concerns. Thank you very much!

---

> ### Author Response · Authors · 2022-08-07
> **Have we addressed your concerns?**
>
> Please let us know if you have more questions. Thank you.

---

### Official Review · Reviewer_6wwh · 2022-07-11

**Rating:** 6
**Confidence:** 4
**Soundness:** 3 good
**Presentation:** 3 good
**Contribution:** 2 fair

**Summary:**

This paper demonstrates a simple baseline range-based 3D detector for multi-frame inputs, which shows the potential way to achieve efficient 3D detection compared with current BEV pipelines. Experiments show the proposed method achieves comparable results with BEV methods such as CenterPoint with a faster speed.

**Questions:**

1. Please see the cons as above.
2. Why FCOS-LiDAR(c128) gets better results than CenterPoint on nuScenes test set and worse results on the val set?
3. As the feature map of each level has to be resized to the original image size, is it necessary to apply FPN to generate multi-level prediction? An ablation study compared with one-level (concatenation or summation) prediction can be provided.
4. Random scale augmentation involved in training might affect all point angles in range view projection. Would it cause object artifacts during projection?


**Ethics Review Area:**

["I don’t know"]

**Limitations:**

Yes, in the conclusion and experiment parts. MRV's performance in multi-frame settings still falls behind BEV's.

**Strengths And Weaknesses:**

Pros:

1. The paper writing is smooth and organized well.

2. The paper comprehensively investigates some basic designs (3D inputs, network backbone, detection head) of range-based 3D detectors and compares the advantages and disadvantages of range-view detectors. Some interesting points such as multi-frame point encoding and modality-wise convolution are proposed and discussed. Overall, it provides a simple baseline approach for RV detectors.

Cons:

The paper lacks a direct comparison with other range-based 3D detectors. Further comparison with RangeDet or RCD or RSN (with the same multi-round range projection inputs) at the level of the operator could further enhance the paper statements, specifically in their speed/runtime comparison. If not, please give the explanations.

---

> ### Author Response · Authors · 2022-08-02
> **Responses to Reviewer 6wwh**
>
> **Q1: Direct comparisons with other range-based 3D detectors.**
>
> None of these RV-based methods releases their full code. This makes it very challenging to compare ours and theirs due to the huge differences in training/testing settings, data augmentation, network architectures, pre-/post-processing etc. In addition, these previous range view detectors are NOT really fully convolutional, being much more cumbersome than ours. For example, both RCD and RSN are two-stage methods. RCD uses a second stage like RCNN to refine the initial box proposals, and RSN only uses the range view for points filtering in the first stage and still relies on BEV in its second stage. RangeDet exploits the Meta-Kernel technique, which dynamically generates the weights of convolutional kernels, hampering the on-device deployment (e.g., TensorRT/ONNX deployment). In contrast, our model is one-stage and only depends on the standard convolutions, thus being significantly simple and easy to deploy on self-driving cars in practice. Moreover, none of the existing range view methods can cope with the multi-frame fusion, which is one of our important contributions and was previously considered intractable in the range view.
>
> We mainly compare our methods with mainstream BEV-based methods such as CenterPoint and PointPillar, showing that an RV-based method compares favourably with these popular BEV-based solutions even in the multi-frame case. These BEV works are used extensively in practice, and thus we think the performance competitive with theirs can demonstrate the effectiveness of our method. Finally, we will release our full code to facilitate the research of RV-based detectors.
>
> **Q2: Different performance compared with CenterPoint on the nuScenes test set and the val set.**
>
> This is because the model size and training setting are different on the test set and the val set. As noted in L337-L339, we only use FCOS-LiDAR(c128) on the test set. The model on the val set is smaller and has only $64$ channels in its detection head. Moreover, for the experiments on the val set, the training/testing settings are strictly controlled to ensure a fair comparison between ours and CenterPoint. For the model on the test set, as noted in L339, we further use the "fade strategy" in [32] during training (i.e., removing the copy-paste data augmentation in the last 5 epochs). This can improve the performance by about 2% mAP. Additionally, the test set results of other methods are directly token from their original papers and there might be other subtle differences in the training/testing process. This is why our method shows better performance than CenterPoint on the test set.
>
> **Q3: The feature map of each level has to be resized to the original image size.**
>
> No, we do NOT resize the feature maps of all levels to the original image size. As noted in L216, only the first level of feature maps has the same size as the original image size, and other levels are down-sampled by powers of $2$, respectively, as in the standard FPN. Thus, FPN is still needed.
>
> **Q4: Does random scale augmentation cause object artifacts?**
>
> Almost not for two reasons. 1) We apply the random scale augmentation globally, i.e., all points in the same point cloud are proportionally scaled by the same scale factor at a time. As a result, this does not alter the azimuth and inclination angles of these points in the spherical coordinates system, and neither do the range view projections of these points. 2) We choose the scale factor in the range from $0.95$ to $1.05$, which only changes the point cloud by a small amount and thus will not cause object artifacts.
>
> **Q5. MRV's performance in multi-frame settings still falls behind BEV's.**
>
> Yes, the multi-frame fusion in range view still is an open question and more efforts are needed. But our RV-based detector is much faster (RV 38.76ms vs. BEV 73.95ms). Ours with a latency similar to BEV-based CenterPoint can achieve a similar result.
>
> |             | Time (ms) | mAP (%) |
> |-------------|-----------|---------|
> | CenterPoint | **74**        | 60.40   |
> | Ours        | 79        | **60.48**   |
>
> Additionally, we would like to highlight that we are the first one to show that range view detectors can also benefit from multi-frame fusion. The difficulty of multi-frame fusion is previously considered to be one of the critical obstacles to the RV pipeline, and none of the previous range view detectors shows any positive results. This impedes the practical application of the RV pipeline that has many unique merits (e.g., avoiding voxelization/sparse convolutions/being lightweight and etc.). Moreover, our work opens up many new possibilities for this promising direction. For example, we enable the RV-based detectors to predict the velocity of the objects with multi-frame fusion (AVE 0.301 vs. AVE 1.08 of single-frame models).
>
>
> Thank you very much for reviewing our work. We sincerely hope you might reconsider the meaning of our work.

---

> > ### Author Response · Authors · 2022-08-07
> > **Have we addressed your concerns?**
> >
> > If you have further questions, please let us know.

---

### Public Comment · ~Siyeong_Lee1 · 2022-12-08
**Code release**

Could you tell us your plans for a code release?

---

### Meta-Review · Area_Chair_8soh · 2022-08-27

**Recommendation:** Accept
**Confidence:** Certain

**Metareview:**

After the rebuttal and discussion two reviewers recommend acceptance, one rejection. In their rebuttal, the authors were able to convincingly resolve all issues raised. Thus the AC sees no reason to reject this paper.

**Award:**

No

---

### Decision · Program_Chairs · 2022-09-14

Accept